# Citizens’ and Farmers’ Framing of ‘Positive Animal Welfare’ and the Implications for Framing Positive Welfare in Communication

**DOI:** 10.3390/ani9040147

**Published:** 2019-04-04

**Authors:** Belinda Vigors

**Affiliations:** Scotland’s Rural College (SRUC), West Mains Road, Edinburgh EH9 3RG, UK; belinda.vigors@sruc.ac.uk

**Keywords:** farmer perception, citizen perception, qualitative research, free elicitation narrative interviews

## Abstract

**Simple Summary:**

The words used to communicate farm animal welfare to non-specialists may be more important than knowledge of welfare itself. Framing research finds that human perception is influenced, not by *what* is said, but by *how* something is said. By increasing the emphasis placed on animals having positive experiences, positive animal welfare changes the framing of farm animal welfare. Yet, we do not know how such framing of animal welfare may influence the perceptions of key animal welfare stakeholders. In response, this study uses qualitative interviews to explore how citizens and farmers frame positive animal welfare and what this means for the effective communication of this concept. This study finds that ‘positive’ evokes associations with ‘negatives’ amongst citizens. This leads them to frame positive animal welfare as animals having ‘positive experiences’ or being ‘free from negative experiences’. Farmers rely more on their existing frames of animal welfare and integrate positive welfare into this. As such, most farmers frame positive welfare as ‘good husbandry’, a smaller number frame it as ‘proactive welfare improvement’, and a small number frame it as an ‘animal’s point of view’. The implications of such internal frames for effectively transferring positive welfare from science to society are further discussed.

**Abstract:**

Human perception can depend on how an individual frames information in thought and how information is framed in communication. For example, framing something positively, instead of negatively, can change an individual’s response. This is of relevance to ‘positive animal welfare’, which places greater emphasis on farm animals being provided with opportunities for positive experiences. However, little is known about how this framing of animal welfare may influence the perception of key animal welfare stakeholders. Through a qualitative interview study with farmers and citizens, undertaken in Scotland, UK, this paper explores what positive animal welfare evokes to these key welfare stakeholders and highlights the implications of such internal frames for effectively communicating positive welfare in society. Results indicate that citizens make sense of positive welfare by contrasting positive and negative aspects of welfare, and thus frame it as animals having ‘positive experiences’ or being ‘free from negative experiences’. Farmers draw from their existing frames of animal welfare to frame positive welfare as ‘good husbandry’, ‘proactive welfare improvement’ or the ‘animal’s point of view’. Implications of such internal frames (e.g., the triggering of ‘negative welfare’ associations by the word ‘positive’) for the effective communication of positive welfare are also presented.

## 1. Introduction

Getting people to act upon, and according to, research evidence is not easy. You do not have to look far within the farm animal welfare literature for examples of key stakeholders (e.g., citizens and farmers) failing to apply evidence-based recommendations in practice (e.g., [1,2]). However, it is often not *what* is said but *how* something is said that determines human perception and action. Decades of framing research has revealed that specific words or phrases can evoke a clear set of associations or ‘ways of viewing the world’ in someone’s mind, and this influences how they understand, evaluate, and thus respond to information [3,4]. Consequently, the way in which information is presented (e.g., the words or phrases used in communication and the salience of positives and negatives) can influence and change how an individual interprets and evaluates something and the choices they make in response [5]. However, this knowledge has not yet received extensive consideration within animal welfare science [6]. This is a noteworthy issue as (i) the internal frames in thought an individual holds can influence how they interpret and respond to research evidence and information and (ii) the way in which research evidence is described and framed in communication may determine the actions animal welfare stakeholders take [5,7], thus affecting the lives of farm animals.

Frames in thought are the internal “cognitive schemas of interpretation, mental filters or ‘mind-sets’” (p.3, [7]) an individual draws from to make sense of and assimilate information. The way in which an individual internally frames something can affect their behaviour by influencing how they evaluate information and determining what they consider most relevant for such evaluation [5]. For instance, an individual framing farm animal welfare in terms of an animal’s affective state will place considerable emphasis on how an animal feels [8], thus influencing how they evaluate the welfare of an animal and how they interpret associated information. However, frames in communication (i.e., the properties of interpersonal communication) can influence what internal frames an individual draws from (or indeed, provide new frames) by making sub-sets of information more or less salient [3,9]. For example, Mellor [10] highlights how the Five Freedoms framework was not formulated to be an absolute standard or represent ideal states, but wider society has interpreted and assimilated the freedoms as being fully achievable. This is likely because ‘freedom’ evokes connotations of ‘absence from’, hence, the salience of ‘freedom’ in the communication of the framework caused individuals to interpret that negative experiences should be eliminated [10]. Therefore, framing in communication has potentially influenced individual interpretation and response to one of the most conspicuous frameworks in animal welfare.

One of the key research areas in the framing literature is whether it is better to accentuate negatives or accentuate positives in communication [11]. This is because a plethora of studies, in numerous contexts, find a *framing effect*, where the alternative framing of objectively equivalent information in positive and negative terms produces divergent responses within individuals [11,12,13,14,15]. This effect of framing in communication is particularly relevant to the newly emergent concept of positive animal welfare. Positive welfare can be broadly understood as a movement to shift animal welfare’s more traditional focus on avoiding the negative aspects of an animal’s life (e.g., pain) towards a focus on enhancing the positive aspects of an animal’s life (e.g., pleasure) [16]. In doing so, positive welfare is arguably changing the ‘framing’ of animal welfare, both in terms of the internal mental frames it intends to evoke (e.g., positive affect rather than negative affect) and in terms of the words, phrases and language used to frame animal welfare in communication (e.g., the opportunity for animals to have a *meaningful* and *pleasurable* life [17]). Although there are differences within the literature on what positive welfare entails (e.g., enhancing positive affect and promoting pleasurable behaviours such as play [18,19]), the shared motivation of this literature is to make clear that animal welfare needs to be updated to include greater consideration of the positive aspects of an animal’s life (e.g., [16,20,21]). Indeed, in most cases, creating a convincing case for positive welfare has relied on distinguishing it from animal welfare’s more traditional emphasis on negatives [10,17].

By emphasising and increasing the salience of positive rather than negative aspects of welfare, positive welfare arguably seeks to re-frame the attributes associated with animal welfare. However, we have little insight into how key stakeholders in society may respond to such attribute framing. This is a pertinent and contemporary issue for the furtherance of positive welfare research, in particular, as indicators and frameworks for assessing positive welfare at the animal level are developing [17,21]. Before positive welfare can be effectively taken forward and promoted in society, there is a timely need to better understand how critical stakeholders, such as farmers and citizens, perceive and interpret positive animal welfare. Moreover, we should be concerned with how their current meaning frames—influenced by their values, beliefs and social interactions—impact their perception of positive welfare, as this may influence the way in which it is adopted and/or applied, and indeed, resisted. Furthermore, understanding how key stakeholders understand and frame positive welfare may prove beneficial for the effective communication and transfer of positive welfare research from science to society (e.g., farmers and citizens). In light of the knowledge of framing and its effect on human behaviour, it is arguably not evidence of positive welfare that may motivate key animal welfare stakeholders to engage in positive welfare behaviours, but *how* that research is communicated. As Blackmore et al. (p.14, [22]) explained, “the presentation of facts or data is rarely sufficient in motivating our concern and action”. Knowing what internal meaning frames such stakeholders draw from to make sense of positive welfare is one key step in ensuring it is communicated appropriately.

In response, this paper seeks to explore and uncover how farmers and citizens frame positive welfare (i.e., frames in thought), and the implications of this for the effective framing of positive welfare in communication. As little is known about how key social actors frame positive welfare and there is a limited consensus within the literature on what should be included within a positive welfare framework [10,16,17,21,23], it is too early to use experimental manipulation of frames to test the impact of positive welfare on behaviour. Moreover, it is likely that positive welfare may be construed and understood in different ways by different people as they draw from personal meaning frames, experiences, values and norms to make sense of it. As such, there is a need to recognise this and employ methods which draw out, exemplify and accept such diverse subjective interpretations of positive welfare in society. Therefore, a qualitative approach is most applicable, as it enables exploration into *how* individuals interpret and make sense of phenomena and, is particularly well suited to the study of topics which are ‘ill-defined’ or ‘poorly understood’ [24]. For this reason, the free association narrative interview method [25] is applied to access farmers’ and citizens’ associations with positive animal welfare and how they frame positive welfare based on these internal points of reference. Uncovering these internal frames, or ‘frames in thought’, is essential for understanding how best to frame positive welfare in communication.

In sum, this paper explores both *how* positive welfare is framed in thought by key animal welfare stakeholders and suggests how *ought* positive welfare be framed in communication by animal welfare science.

## 2. Methodology

This article draws from qualitative interview data collected as part of a research study, undertaken in Scotland, UK, to explore experts’ (farmers) and lay-persons’ (citizens) perspectives on positive animal welfare. The study explored a number of factors including how livestock farmers and citizens freely interpret and ‘frame’ positive animal welfare, and the factors underlying and influencing this framing and the extent to which aspects of positive welfare are evident in their current perceptions of animal welfare. This present article primarily focuses on the outputs relating to how positive welfare is framed by farmers and citizens and the implications of this for the effective communication of positive welfare.

### 2.1. Method

The interview protocol was designed according to the ‘free association narrative interview method’ described by Hollway and Jefferson [25]. This is an open approach to interviewing, which combines the story-telling role of narratives with the psychoanalytical principle of free association:
“Which assumes that unconscious connections will be revealed through the link that people make if they are free to structure their own narratives”.[25]
Unlike most interview-based methods, which assume that participants can ‘tell it like it is’, it posits that interviewees are unlikely to understand the question in the same way as the researcher [25]. As such, a critical aspect of interview design was ensuring participants had space to recount their understanding of the target phenomenon, free from any *a priori* research assumptions affecting data collection (e.g., by asking narrow or closed questions). Although there is no singular way to conduct narrative interviews, the generally recommended approach is to first pose a primary narrative inducing question, where the researcher listens passively (e.g., does not offer any prompts) whilst actively taking notes on key points generated by the participant [26]. Once it is clear the participant has completed the recounting of this main narrative, the researcher may then use these notes to ask further questions and probe particular points [26]. In this regard, a particular strength of this interview method is that participants are not restricted to the researcher’s agenda or the questions the researcher asks. Rather, it encourages participants to tell their story and to take ownership of how that story is told [25].

In order to explore the associations the term ‘positive animal welfare’ brought to participants’ minds and how this influenced their framing and interpretation of it, the interview design aimed to encourage participants to ‘think-out-loud’ about positive welfare. For this reason, a singular free elicitation question: “Positive animal welfare—can you tell me about what comes to mind when you hear that?” was posed. No prior explanation of positive welfare was provided, and no additional prompts were given after this main narration question was asked. This left the participant free to set the agenda of discussion and construct their own narrative around positive animal welfare. By doing so, the researcher was provided with a lens into what factors the participants associated positive welfare with, what they drew from to make sense of it and their overarching frame of positive welfare. Once this main narration was completed, further details on points raised by participants could be asked by the researcher (e.g., can you give an example of what you mean by ‘natural’). Through constructing and recounting the narratives of what positive welfare ‘brings to mind’, participants revealed the experiences and associations most relevant to their framing of positive welfare and the importance of the subjective meanings they attach to it [26]. This approach produced rich and diverse discussions across participants and provided insights into the meaning frames they drew from to make sense of and assimilate positive welfare.

### 2.2. Sample

Farmers and citizens were the chosen case studies as they represent two key stakeholder groups in animal welfare, with farmers directly impacting animal welfare through their behaviour and management practices and citizens indirectly affecting welfare through their preferences [27,28]. The study used a purposive heterogeneous sampling approach to select an information-rich and diverse range of research participants [29,30]. As positive welfare is a relatively new concept, this approach enabled as much insight as possible to be gained from a variety of different farming systems and perspectives and a variety of different people in society.

Farmers were recruited through various means, including advertisement on social media, referral by farm advisers and other industry stakeholders, attendance at farmer events (e.g., monitor farm meetings) and by directly contacting individual farmers via email. Face-to-face interviews were completed with 13 dairy farmers, 10 beef and sheep farmers, 2 free range egg producers, 2 mixed farmers (i.e., pig, poultry, beef and sheep) and 1 large pig farmer (total n = 28). These interviews took place in the homes of individual farmers in Scotland between March and September 2018 and lasted, on average, 55 minutes. The majority were males (86%), while a small number were females (11%) (one participant preferred not to indicate their gender). In reflection of the heterogeneous sampling approach, a variety of farming systems were evident within each sector (i.e., zero-grazed or grazed in dairy, indoor-wintered or outdoor-wintered in the beef and sheep sector, and organic or non-organic in the poultry sector). Further demographic details, specific to each participating farmer, can be found in Table A1 in Appendix A. 

Citizens were initially recruited through advertisement on social media outlets and in public spaces (e.g., libraries, local shops and community centres) in large urban, small towns and accessible rural areas in Scotland [31]. However, this resulted in a very poor response rate with only five responses and three individuals volunteering to participate. As such, a further 12 citizen participants were recruited using the market research company ‘Testing-Time’, resulting in 15 citizen interviews in total. Citizen interviews took place in Scotland between April and September 2018. Three interviews were completed face-to-face, while the remaining took place over Skype at the preference of participants. The average interview duration was 50 minutes. In total, 60% of participants (n = 9) were females and 40% (n = 6) were males. The majority of the sample lived in large urban areas (n = 11) with the remaining evenly split between small towns (n = 4) and accessible rural areas (n = 4). Reflecting heterogeneous sampling, citizen participants differed in their dietary preferences with six stating they consumed meat, six indicating they have reduced meat consumption in the past year, while two participants were vegetarian and one vegan. Participants also differed in their experiences of farming—four participants came from a farming background (i.e., worked on farms in the past or had family members who farmed); seven noted they had visited farms (i.e., on educational trips); and four stated they had no experience of livestock farming at all. The specific demographic details of each citizen can be found in Table A2 in Appendix A.

The research was approved, before commencement, by the SRUC Social Science Research Ethics Committee. All interviews were audio-recorded with participant permission, except in the case of one farmer interview (who preferred not to be recorded) where detailed notes were taken instead. As with any qualitative research, the aim was not to determine outputs that could be generalised to a population but to reveal meaning and understanding, in the context, of how different individuals framed positive animal welfare. As such, once a saturation point had been reached (i.e., no additional data, from which to develop properties of a category, emerged [32]), no further participants were sought.

### 2.3. Data Analysis

Each recorded interview was transcribed in full and entered into the qualitative data analysis package MaxQDA to assist analysis. To distance the exploration of farmers’ and citizens’ framing of positive welfare from any *a priori* definitions of positive welfare, the constant comparison method was used to analyse the data [32]. This entailed reading each interview and categorising or ‘coding’ sections of text according to the nature of the points raised. For example, codes emerging from farmers’ narratives in response to ‘what positive welfare brings to mind’ included healthy animals, improving welfare, stress reduction and happy animals (amongst others). Once this was completed, coded sections were compared between interviews and related codes, and comments were grouped to form overarching key themes. This ensured the generation of insights that were grounded in the data [33]. These key, overarching themes then formed the basis for uncovering how farmers and citizens framed positive welfare, as outlined in the subsequent sections.

## 3. Findings

This section describes how citizens and farmers interpret and make sense of positive animal welfare and the different ways in which citizens and farmers frame positive welfare. The first sub-section illustrates the sense-making process evident in participants’ free association narratives before the subsequent two sections detail how citizens and farmers frame positive welfare and the themes characterising each frame. Throughout, participants’ narratives are directly drawn from and presented to ensure a transparent link between their views and the researcher’s interpretation of these views. The data is analysed in more detail in the subsequent discussion section.

### 3.1. Participants’ Sense-Making of Positive Animal Welfare

In seeking to understand how farmers and citizens made sense of and gave meaning to positive welfare, it is important to first note that neither group had heard of the term ‘positive animal welfare’ before. As one citizen explained:
“When you said positive animal welfare, I thought that’s a strange term; I’ve never heard that before”.(Citizen 2)
Similarly, farmers often reflected that the interview had been:
“The first time I’ve ever heard that term used”.(Dairy 6)

As such, in seeking to give meaning to and respond to a term previously unknown to them, many participants focused on the word ‘positive’, as this represented what made this term strange, relative to their existing constructs of animal welfare.

Amongst citizens, it was evident that the term ‘positive’ triggered associations with ‘negative’; participating citizens rarely made sense of positive animal welfare *without* using a comparative opposite. For example:
“So the welfare should be **not** to exploit them, **not** to obviously harm the animals”.(Citizen 4)
“Have a good quality of life, for me, that means sort of as natural a life as possible, you know, young **not** being taken away from parents, having access to the outdoors and the fresh air and all that sort of thing, and **not** being fed anything artificial”.(Citizen 6)
“I think it is the animal’s experiences, positive experiences rather than like **negative**”.(Citizen 7)
“Just a nice kind of hill for them, like fenced around, so **nothing** can get to them that is going to **hurt** them, a few sheep in there; **not** crowded at all”.(Citizen 9)
“It’s the exact sort of **opposite** of that imagery; I’ve got the horror image in my head right now and then the other image is the exact **opposite**”.(Citizen 3)
As highlighted by the comments in bold (emphasis added), when de-constructing ‘positive animal welfare’, citizens made sense of it by juxtaposing positive aspects (e.g., ‘a natural life’) with negative aspects (e.g., no harm). In this way, many citizens deduced that positive welfare was something which did not involve such negatives. Indeed, for some citizens, this comparative association between a ‘positive’ and a ‘negative’ of welfare was explicit:
“Positive animal welfare? Well, what would be **negative animal welfare**? That would be the question”.(Citizen 2)

Overall, citizens’ free association narratives revealed that the term ‘positive’ engendered potentially automatic associations with ‘negative’. As will be shown in the subsequent section, this influenced how they framed ‘positive animal welfare’ overall.

Farmers demonstrated a much more nuanced approach in their sense-making of positive welfare. Some focused directly on and engaged in semantic analysis of the word ‘positive’ as a means to understand the whole term ‘positive animal welfare’:
“Positive means to add. So therefore, you are adding above the baseline”.(Dairy 10)
“I suppose positive: higher than you need to be, crossing a bar”.(Beef & Sheep 10)
However, in most cases, farmers drew from their personal constructs of animal welfare to make sense of ‘positive animal welfare’ as a whole. In other words, unlike the prior examples which deduced positive welfare by first thinking what ‘positive’ means *per se*, the majority of farmers’ sense-making centred on the deduction of ‘what are the positive aspects or outputs of my current animal welfare practices?’ For some farmers, although not quite as overt as citizens’ juxtaposition of ‘positive’ versus ‘negative’, positive welfare was surmised to be a combination of preventing negatives and enabling positives:
“Positive welfare…. It’s trying to prevent anything bad from happening to them, you’re trying to maintain comfort, you’re trying to maintain health and well-being”.(Dairy 6)
Nevertheless, the majority of farmers focused on their ‘ideals’ of welfare to give meaning to positive welfare:
“Keeping them well-fed, bedded, watered and their health looked after…. So, they’re cared for and looked after to the best of our abilities; if they’re ill, or there’s a problem, that you look after them”.(Beef & Sheep 7)
Interestingly, pig and poultry producers’ sense-making of positive welfare centred around the five freedoms:
“You know, if you want to be scientific about it you’ve got the five freedoms, which, yeah, I wouldn’t disagree with. You know, if you go through those five freedoms that is basically what you are trying to give the animal”.(Organic Free Range Egg 1)
“You’ve got something called the six freedoms I think it is. And it is to do with the freedom from fear, freedom from hunger, freedom from thirst, freedom from stress, freedom to exhibit natural behaviours and there is one more that I can’t think of. And that is basically it. As long as pigs have all of those, then to me that is good enough…. Humans would be the same, pigs are the same, sheep would be the same. As long as they can do all of those things, they should be relatively happy, I guess would be the term”.(Pig 1)

Overall, it was evident that citizens and farmers focus on their associations with the word ‘positive’ to make sense of positive animal welfare. For citizens this conjured up associations with ‘negative’, triggering an overt comparison of welfare opposites (e.g., negatives versus positives, or the ‘dos and don’ts’) to make sense of and distinguish positive welfare. Farmers, on the other hand, drew from their semantic associations of the word positive (e.g., ‘to add’, ‘raise the bar’) and their existing constructions of the negative welfare aspects they sought to reduce and the positive aspects they sought to provide. Moreover, they did not usually separate negative and positive but rather saw both as mutually inclusive aspects of overall good welfare.

### 3.2. Citizens’ Framing of Positive Animal Welfare

Citizens clearly framed positive welfare in one of two ways: (1) positive welfare as ‘positive experiences’ for farm animals, or (2) positive welfare as being ‘free from negative experiences’. Of the four participants who stated they had a farming background, three of them demonstrated a ‘free from negatives’ frame, while one framed positive welfare as ‘positive experiences’. Amongst the seven participants who expressed they had experienced livestock farms (e.g., through educational visits), five framed positive welfare as ‘positive experiences’ and two as ‘free from negatives’. The four participants who stated they had no experience of livestock farming were evenly split between both frames, with two demonstrating the ‘positive experiences’ frame and two demonstrating the ‘free from negative experiences’ frame. The vegan participant framed welfare as ‘positive experiences’, while one vegetarian demonstrated the ‘positive experiences’ frame and the other the ‘free from negatives’ frame. Similarly, the six participants who stated they had been actively reducing their meat consumption were evenly split between the two frames.

#### 3.2.1. Positive Experiences

The ‘positive experiences’ frame was made evident by the emphasis individual citizens placed on farm animals having opportunities for positive experiences. To present this mental representation of positive welfare, participants focused on and made salient several factors which they considered provided positive experiences to farm animals. These included (i) a natural–outdoor environment:
“Just that animals are being cared for in the best way for them, for me that conjures up ideas of the most natural…having access to the outdoors and fresh air and all that sort of thing”.(Citizen 6)
which was further seen as a means to ensure animals had desired (ii) space:
“I think it is more animals with free space in large fields”;(Citizen 8)
which was further inextricably linked as a way in which animals could be provided with opportunities to exert (iii) autonomy over their environment:
“Just being outside in the fresh air and having the sun, I think these things are good for humans so I think that would extend to animals as well. Freedom to roam, trying to have as normal a life as possible, what a cow would do day to day, allowing them the space and freedom to do that”.(Citizen 2)
“Positive, a positive experience for an animal…. Maybe like in the case of pigs, a little mud bath or something like that, something that they can choose to do, they are not being forced to do it. They don’t need to do it to survive. They are not being forced to do it by a farmer”.(Citizen 9)

In addition, the representation of a traditional or pastoral farm as being the best way to create a (iv) positive human–animal relationship between farmer and farm animal was further made salient by participants within this frame:
“The first thing that comes to mind is the happy farmer and happy animals….I think it’s the love between them and if they are showing love to the animals it is like both sides are happy…so I guess in a broader sense it is just a happier environment, more than just the animal. I feel like it is a happy environment on both ends, where animals are being looked after by people who care for the animals”.(Citizen 4)

These factors were not exclusive but coalesced to create this frame, or mental representation, of positive animal welfare. Specifically, a natural–outdoor environment was framed as a prerequisite for providing animals with the desired space for them to exert autonomy within their environment, which was further supported and enabled by a positive or caring human–animal relationship.
“So I think positive animal welfare to me, is not just open fields and all that, which it is, that is usually the first image that comes to mind, but it is also one of these small little farms where they have a quality life, and it is a lot smaller and you know someone who knows every single one of the cows and has probably even named them, I don’t know. So, you know, there is a bond between the farmer say, with the animals, and the animals with the farmer, for sure. And there is love, that probably does [matter], it does you know”.(Citizen 1)

Thus, participants who framed positive animal welfare as animals having the opportunity for ‘positive experiences’ placed emphasis not just on the prevention of negatives but on the factors which they considered gave animals a positive life namely (i) access to a natural–outdoor environment, (ii) space, (iii) autonomy and (iv) a positive human–animal relationship.

#### 3.2.2. Free from Negative Experiences

The ‘free from negatives’ frame differed from the ‘positive experiences’ frame in that little to no emphasis was placed on the provision of positive experiences. Rather, the emphasis here was that not experiencing any negatives would provide an animal with a positive life. In other words, positive welfare was framed as the absence or elimination of negative experiences.

In this regard, citizens made salient the importance of ensuring (i) no harm:
“Treat them how you want to be treated….so the welfare should be not to exploit them, not to obviously harm animals.… So before it was obviously killed and what not, you did the best for it, you didn’t kind of treat it bad or negatively, you didn’t give it a bad life”(Citizen 5)
“What I think it means is that animals are treated in the way they should be treated and not tortured in a farm let’s say. So the picture in my head is animal’s free out in nature or the other scenario would be the animals that are treated, they may be not in nature, but at least they are treated properly and they are not being tortured”;(Citizen 14)
(ii) eliminating negative affect:
“Because I think not letting them feel fear. I’m sure around the world some animals have been really abused. So like having that fear that they could have…. That is definitely not a positive experience”;(Citizen 7)
and (iii) preventing health issues:
“I guess, to me, more kind of like their health, so they are not kept in a way that is going to make them ill or that is going to be detrimental to them”.(Citizen 11)

Thus, participants demonstrating this frame interpreted the prevention of harms, negative affect and health issues as being a positive thing, in and of itself. As such, positive animal welfare was framed as animals being free from negative experiences.

### 3.3. Farmer Framing of Positive Animal Welfare

Amongst farmers’ responses to positive animal welfare, three frames were evident: positive welfare as (i) good husbandry; (ii) proactive welfare improvement; and (iii) the animal’s point of view. ’Good husbandry’ was the predominant frame with seven of the dairy farmers, six of the beef and sheep farmers, and all of the pig and poultry farmers demonstrating this frame. A smaller number—five dairy farmers and four beef and sheep farmers—framed positive welfare as ‘proactive welfare improvement’. The remaining two mixed farmers and one dairy farmer framed positive welfare as considering the ‘animal’s point of view’.

Interestingly, across all participating farmers, a pre-existing frame of animal welfare as ‘happy–healthy–productive’ animals was persistent and transcended across all three frames of positive animal welfare. Specifically, each farmer framed the productivity and performance of their animals as the key ‘objective’ indicator that their animals were well cared for. However, productivity and performance were given meaning by the emphasis placed on ‘happy–healthy’ animals. Namely, ‘productiveness’ came from an interaction between ‘happy’ and ‘healthy’, whereby healthy animals were framed as happy animals and an unhappy animal was framed as being unlikely to be healthy. Farmers used and applied this pre-existent frame of animal welfare to affirm and demonstrate that the welfare of their animals was positive. In other words, a productive farm indicates well cared for animals (i.e., happy and healthy) and therefore, a productive farm shows that animals are being positively cared for and managed:
“So I suppose the weigh scales will tell you how happy they are…because if they are happy they are putting on weight, which is what I am trying to do”.(Beef and sheep 6)
“And animal welfare and health and all those things around about is our priority. Because we want to create a, we want to create the best environment for our livestock. Because if they are happy, then they are going to produce more milk, be healthier and make our life a lot easier”.(Dairy 11)
“Well why I consider my hens have a good life is, well it’s very simply because we, whilst my hens work for me, everything that we do here is all about hen welfare, and about trying to get, and it’s a terrible thing to say, but you know I told you earlier, profitability and output from animals is very directly linked to their health and welfare”.(Free Range Egg 2)

Overall, the farmers group demonstrated some differences between participants in terms of how they frame positive welfare and therefore interpret it, but each participating farmer drew from the same overarching animal welfare frame of ‘happy–healthy–productive’ animals. This, as is shown in the subsequent sections, impacts the way in which each positive welfare frame is constructed.

#### 3.3.1. Good Husbandry

The ‘Good Husbandry’ frame was the most prevalent frame of positive welfare. This frame was made clear by the emphasis these individuals placed on factors they considered meaningful for providing the best possible care or, good husbandry, for their animals. This included a strong emphasis on (i) health:
“Yes, performance. I suppose it goes back to measure, manage, measure. I look at welfare as a sort of health issue, and I suppose if they’re healthy, they perform better”;(Beef and Sheep 7)
which, as previously discussed, was closely tied with and considered a prerequisite to having (ii) happy animals:
“But we would hope that we are doing everything and more to, if you create the best environment that you can create, within reason, and can keep them as happy and healthy as you can”(Dairy 11)
“Just treating your cow, treating them right, treating them well. As I say you want just happiness. It does sound silly, but happy cow happy herd really. Positive animal welfare, well it’s you just want things to be good. They’re never going to be perfect but you can aim for it”.(Dairy 9)
As alluded to in the prior narrative, farmers also stressed their own role within the ‘good husbandry’ frame, where they emphasised the importance of (iii) doing the best they can, for their animals:
“I suppose it’s offering the best for them. Like for me, it’s having a shed that you’d be happy if anybody came in and took a photograph over the gate in the winter time and the same in your fields. So nothing that you wouldn’t want somebody to go and see, whether it’s slat mats; that they’re well stocked and they’re clean”.(Beef and Sheep 3)
Ensuring (iv) resource needs are well met:
“They don’t have any further needs, they have their food there, they have got plenty of food, they’ve got a nice bed, they’ve got water, and they are content”;(Dairy 11)
and reducing (v) stress:
“Lack of illness, just that lack of stress really is my take on it”.(Beef and Sheep 5)
“So for me, welfare is actually quieter cattle, far easier to work with…they’re not stressed. To me, that’s more how I’d say I treat welfare”(Beef and Sheep 4)
were also presented as key themes within the ‘good husbandry’ frame of positive welfare.

As illustrated by the above narratives, each theme is not discussed in isolation but are interconnected aspects of farmers’ interpretation of positive animal welfare. Namely, a happy animal is a healthy animal, which is what farmers want to achieve by providing their animal with the ‘best’, which further requires them to minimise stress, so the animal can, in turn, be healthy and happy and the farmer, simultaneously, can know they are doing their best. Thus, there is a cyclical relationship between each of the above-mentioned factors which, when taken as a whole, forms the positive animal welfare as ‘good husbandry’ frame. In short, farmers used these factors to construct and convey their interpretation of:
*“Positive animal welfare…* [as being] *about looking after your animals in a good manner and good welfare”.*(Beef and Sheep 7)

Thus, it is evident how the ‘happy–healthy–productive’ frame transcends and influences farmers’ interpretation of positive animal welfare here. However, as will be made evident by the subsequent sections, the ‘good husbandry’ frame is distinct in that farmers here did not consider positive welfare to be much different to or anything beyond their current practices.

#### 3.3.2. Proactive Welfare Improvement

Those who framed positive welfare as ‘proactive welfare improvement’ saw it as something which involved going beyond the standard levels of animal welfare. However, this did not mean their overall frame of welfare was completely distinct from those discussed in the previous section. Possibly due to the ubiquitous nature of the ‘happy–healthy–productive’ frame, many farmers demonstrating the ‘proactive welfare improvement’ frame also emphasised ‘good husbandry’ themes such as minimising stress:
“Trying to make them live comfortably for as long as they can, trying to be stress-free, pain-free, injury-free”,(Dairy 6)
ensuring resource needs are met:
“Just providing enough, providing all the things they need, the food and water, the other animals to keep them company and the disease, the treatment of disease whenever possible. Things like clipping feet and trimming feet when they need it, just the things that you need to do to keep them content”,(Beef and Sheep 10)
and the importance of health and happiness:
“It means that the animals should be happy, contented and healthy”.(Dairy 1)

However, unlike the previously discussed group, farmers here differed in that they interpreted positive animal welfare to mean going beyond the norm and doing something to actively improve or make changes to their current practices. In particular, they emphasised that:
“It is about being proactive”(Beef and Sheep 2)
where ‘being proactive’ meant doing things to improve welfare beyond basic standards or norms:
“I guess taking what we’ve got and improving on it. Accepting where we are and trying to get better”,(Beef and Sheep 9)
“And then that is your farm, you have got this slightly higher, you are doing better than you have to, yeah high, I suppose positive: higher than you need to be, crossing a bar, that is a definite again”.(Beef and Sheep 10)
“I don’t think you can be successful without exhibiting some positive animal welfare. With that kind of thing, I am sure there are aspects that every farm can improve and most farmers will strive to improve them…. I think we are conscious of, well I am conscious of anyway, the next thing that you should be improving is the thing that you feel least comfortable showing to somebody”.(Dairy 6)

Interestingly, farmers within this frame often anchored this interpretation of positive welfare in their own personal motivations, where they emphasised the personal value they place on actively and continuously seeking to improve or do things differently:
“If you think you have already done all you can do, and you can’t get any better, then you will never get any better. But if you have some kind of goal…There is always a goal and an achievement that you should look to and if that means, if that is positive animal welfare then that is positive thinking. You have to think that you could improve somewhere, something, otherwise what are you working towards? You will never sustain yourself if you reach a level and then decide that you’ve reached. Like if we thought we’d bred the best cow we were ever going to breed, then we might as well give up now because we are never going to breed another one. That is the way I see it…. [Overall]….It sounds to me like it’s something that is going to be a benefit to us. So if it’s positive animal welfare it is positive for the animal but it is also positive for our system as well. That is essentially what it means”.(Dairy 8)

In sum, farmers who demonstrated a ‘proactive welfare improvement’ frame felt positive welfare was about more than just the appropriate provision of resource needs or the maintenance of health (i.e., good husbandry); they saw positive welfare as actively thinking about how to improve welfare or go beyond basic welfare standards. In many instances, this was closely related to their personal, perhaps intrinsic, motivations and self-image as a proactive individual.

#### 3.3.3. Animal’s Point of View

A small number of farmers, just three in total, framed positive welfare as considering the ‘animal’s point of view’ when making decisions. Again, it was not that these individuals did not emphasise the importance of ‘good husbandry’ or ‘happy–healthy–productive’ animals, it was that they made salient additional factors which set them apart from those discussed in the previous two sections. Specifically, they continuously returned to and emphasised how, when making decisions, they tried to think about it from the animal’s point of view or take into consideration an animal’s preferences:
“Positive animal welfare. I think it is putting, the way I see that is you would look at the cow’s perspective on life rather than yours. What would the cow want out of life rather than what you would want the cow to have? What would the cow want to do?”.(Dairy 5)
“And if animals, I mean you can’t tell me that when you have lambs at about 4-6 weeks, and they start jumping up on the silage bales, they are not playing. You know, they are not jumping up on the silage bales to get food. Or to have sex, which is what all the psychologists seem to think motivates all of us…. You know, they are playing, they run around and they form a gang… so, farms need to be places where they can do that”.(Mixed 1)
“Positive animal welfare…. It’s giving all animals, no matter what their species, a positive life. Positive environment. Just actively constantly thinking what’s right for this animal in this system. Within this production system, what’s the positive?”.(Mixed 2)

Interestingly, when discussing ‘positive animal welfare’ and their association of it with an ‘animal’s point of view’, they often juxtaposed it against some of the more basic welfare provisions (e.g., resource needs) to emphasise how, even when animals are appropriately provided for, they still felt it was important to consider their preferences and point of view. For example, dairy farmer 5 continued from their above narrative with:
“What would the cow want to do? I feel as if we underestimate that. We think we know, and we think we can, because it is an animal, we herd it and we tell it what to do and all the rest of it. Might be right, but I think we should still look at what the cow wants. I know that my cows, and again…we’ve got mats and everything else, scraper systems, and washed down every day, everything is washed every day and clean and the cows are clean. But the cow is still, and you know they’ve got lovely high quality, again prize-winning silage and a TMR mix that has all been designed by nutritionists and all the rest, and analysed and sampled and footered about with, and the length of the straw that we chop in to it, it’s got to the right length of it, it’s all there. But they are still right keen to go out of the house, so you’ve got to let the cow go out. You’ve got to let it go out…because that cow will reward you by you giving it it’s choice”.

Thus, farmers within this frame emphasise that even with the best quality of care, or the best provision of resource needs, or the best management systems, the animal won’t have a positive life if their preferences and point of view are not taken into consideration. This is what differentiates this group from the previously discussed frames.

### 3.4. Farmers’ Response to Positive Animal Welfare

Farmers, in addition, volunteered their opinion on positive animal welfare. Here, the former finding regarding citizens’ contrasting of positive and negative is particularly noteworthy considering the concerns many farmers expressed about the term ‘positive animal welfare’:
“I don’t know, the positive bit doesn’t seem right. The word positive doesn’t seem right. The positive, the first thing you think of when someone says positive is negative. I would say. I would say by emphasising that, you would make people think; well there is positive but that means there must be negative. What word would you put instead? I don’t know, I can’t think of another word, but to me, that, whenever you say positive you invite the comparison towards negative”.(Beef and Sheep 10)
Consequently, many farmers were perturbed, and understandably defensive, about what ‘positive animal welfare’, due to its associations with negative, could engender in wider society:
“Why would they want to do that? Is it that they think that we are not having a positive welfare outlook?”.(Beef and Sheep 1)
“When I hear the term positive welfare, ahh, I suppose I start to be fearful of that term; positive animal welfare, because I think it sort of nearly, we get painted, farmers get pained in an unfair light most of the time”.(Beef and Sheep 6)
“My only concern would be…would it be perceived as, well why are you promoting that? Why is that not happening already? Kind of a thing”.(Dairy 3)

However, in addition to concerns that positive welfare evokes the idea that negative welfare exists, some farmers welcomed positive welfare when they gave meaning to it as something which would help promote a positive image of farming in society:
“I think positive animal welfare, I think that’s what we need to promote to a wider public; we are going beyond the, beyond what we do, beyond what the public perceive us to be doing”;(Dairy 12)
or something which would differentiate them from others:
“It would be nice to think that we could get recognised more for the extra effort and that. And we’re not saying that other people don’t go that extra mile…. But how do we get recognised between trying to do the best we can do, and also maybe somebody further down the road that is just getting by and probably could do a lot more to make that a nice place, a nicer environment”;(Dairy 11)
or as something that would enable a desired re-framing of welfare within the farming community:
“Relief—because for too long the word welfare is something that as farmers we have shied away from. We say ’Oh, there’s none of that here, I know we’ve got it but’. Welfare is the chronically lame sheep, it’s a dead lamb that has never been seen and has never been lifted. It is something that you don’t want to see. It is something that you just don’t want to know about and it’s the little guy with a clipboard at the market that is kind of peering over the pen and gives you a look. And that is what welfare has meant. Whereas that is not what it is at all, that is cruelty. Welfare is about the things that actually make us profitable as a business. It’s about the things that help us stay in business…. I think it is a bit of a stumbling block for the industry… we need to get our heads over that. And I think it is a shame that welfare has become about something that you want to hide because somebody might pick on you for having a lame sheep. And we all have lame sheep”.(Beef and Sheep 9)

Thus, farmers demonstrated diverse opinions about positive welfare, with some worried by how it may be perceived by wider society and others interested by its potential to communicate a positive message, differentiate their products, or promote a change in mindset.

## 4. Discussion

This paper set out to explore how positive animal welfare was framed by farmers and citizens, and the implications for the effective framing of positive welfare in communication. Findings reveal that both groups relied heavily on their interpretation of the word ‘positive’ and the associations it brought to mind to make sense of ‘positive animal welfare’ as a whole. Thus, a framing effect is arguably evident; the inclusion of the word ‘positive’ shaped and influenced how these key stakeholder groups made sense of positive welfare and the attributes of welfare they emphasised and gave importance to.

The reliance of citizens on the word ‘positive’ for sense-making may be an outcome of the limited knowledge and first-hand experience they have of farm animal welfare [34]. However, the automatic associations with ‘negative welfare’, ‘positive animal welfare’ evoked may be a natural process of linguistic inference. Within the study of linguistics, polarity is considered a key feature of adjectives, meaning descriptive words have a comparative opposite [35]. A positive adjective signals to a person that a negative form does exist but in this instance, it is not negative [35]. In the context of positive animal welfare, this manifested in citizens’ apparent need to think through what is *not* positive (i.e., what is negative) to make sense of positive animal welfare overall. Regardless of the underlying reason, the positive–negative associations brought to mind by the term ‘positive animal welfare’ was central to how citizens constructed their framing of positive welfare.

Beyond the positive–negative comparisons made by citizens, citizen experience of farming may account for their framing of positive welfare. The ‘free from negatives’ frame was mainly demonstrated by participants with a farming background. Similarly, Nijland et al. [6] found that individuals from rural areas are more likely to frame farming in ‘realistic’ terms with a focus on the extrinsic value of animals (e.g., production). Thus, those with a farming background, due to their socialisation within the welfare norms and values of farming communities, may prioritise the elimination of negative factors affecting welfare (e.g., health issues), as many farmers do [36,37]. Interestingly, nearly all participants who had visited farms (i.e., on educational trips) demonstrated the ‘positive experiences’ frame. Ventura et al. [38] noted that farm visits can reduce citizens’ concerns about the provision of basic resource needs, such as access to food and water, but increase their concerns about animal-based provisions, such as pasture access and cow–calf separation. Thus, visiting and experiencing farms may have led such participants to place greater emphasis on the provision of positive experiences for animals, rather than the prevention of negative experiences. In addition, this finding may further support arguments within the literature that reducing the knowledge distance between farmer and consumer (e.g., through farm visits or educational product information) helps improve citizens’ understanding of farming practices and animal welfare [39,40]. More generally, the majority ‘positive experiences’ frame is consistent with findings that citizens tend to focus more on the positive aspects of an animal’s life [41].

To some extent, citizens’ interpretation and framing of positive welfare could be argued as being reflective of the scientific literature on positive animal welfare. For instance, participants within the ‘positive experiences’ frame freely elicited some of the positive welfare opportunities noted within the literature, such as animal autonomy [42] and positive human–animal relations [17,42]. However, positive welfare so framed brought to citizens’ minds images of traditional, small-farms with animals living a ‘natural’ life outdoors. This is consistent with previous research on citizens’ attitude to animal welfare (e.g., [36,37,43]) and indicates that their understanding of what positive welfare entails derives from personal values that small-scale extensive farming is the best for animals’ quality of life [44]. Interestingly, participating citizens, particularly those in the ‘free from negatives’ frame, emphasise an *elimination* of negative experiences. This is evidenced in their repeated use of overt negatives, such as ‘no’ and ‘not’. However, despite its emphasis on the positives in an animal’s life, the positive welfare literature does not suggest that the prevention of negative aspects of welfare is no longer important, nor does it suggest these should or can be eliminated [10]. Rather it calls for a minimisation of the negatives and an enhancement of the positives, whereby promotion of the latter may incidentally produce the former [17]. Thus, the addition of ‘positive’ to ‘animal welfare’ may lead to similar issues associated with the Five Freedoms framework [10]; the language and words used lead people to deduce that negative aspects can and should be eliminated. As such, the phrasing of positive welfare appeared to influence citizens’ expectations of what it would entail (e.g., elimination of negatives and promotion of positives).

Farmers’ framing of positive animal welfare is ‘fuzzier’ than that of citizens; there were overlaps and similarities across the three farmer frames. This is likely because of the pre-existing norms and values farmers associate with animal welfare, as evidenced in the ‘happy–healthy–productive’ frame transcending between individual farmers. Indeed, the ‘good husbandry’ frame has similar attributes to the ‘happy–healthy–productive’ frame and the existing literature (e.g., [37,45]). For example, Skarstad et al. [46] similarly found that farmers emphasise the adequate provision of resource needs (i.e., food and water), keeping animals healthy and ‘generally taking good care of the animals’ (p. 80). However, farmers in the ‘good husbandry’ group discussed these themes directly in the context of positive animal welfare, indicating they assimilate positive welfare as being comparable to what they already strive to do (e.g., provide animals with good care). As such, the majority of farmers, when posed with the concept of positive welfare, integrated it into their existing frames of welfare and used it to reinforce their ideals as good farmers who looked after their animals ‘positively’.

The superordinate nature of the ‘happy–healthy–productive’ frame and the relatively large number of farmers demonstrating the ‘good husbandry’ frame indicates the importance of pre-existing social norms and values in this context; these were the internal meaning frames farmers drew from to make sense of positive animal welfare. However, despite the strength of these pre-existing frames, it is notable that three different frames of positive welfare emerged. This may be a result of differences in personal values and motivational drivers between farmers. Indeed, farmers in the ‘proactive welfare improvement’ frame often discussed their own personal motivations in this context, where they emphasised the value they placed on seeking ways to continuously improve their current practices. Indeed, Hansson et al. [47] found that the emphasis farmers placed on different economic use or non-use values of animal welfare was strongly related to their personal values (as opposed to personality traits). Notably, they found that animal-centred farmers gave the most importance to non-use values of animal welfare (i.e., economic value not derived from the direct use of the animal). Thus, the ‘animal’s point of view’ frame of positive welfare was potentially shaped by the personal predilection of these farmers to give priority to animals’ wants and desires, regardless of the direct economic value of such acts. Different motivational orientations may thus account for why farmers demonstrated different frames of positive welfare. For instance, when investigating the motivational drivers of European farmers, Baur et al. [48] found that most farmers were more conservative than the general population and tended to be less open to change. This study’s finding that most farmers have a ‘good husbandry’ frame may be a reflection of such conservative motivational orientations, where positive animal welfare was conservatively integrated into existing welfare norms. Nevertheless, the remaining farmers presented very different values and motivations. Further research in this area would thus be of benefit, to explore how such individual motivations and values may affect the way farmers assimilate and respond to positive animal welfare.

The key point is that frames—both in thought and in communication—provide an individual with a reference point for what is important [49]. Amongst citizens, it was evident that the communication of ‘positive’ and its automatic association with ‘negative’ was central to their understanding of positive welfare. In addition, their internal frames of reference about animal welfare influenced what they perceived positive welfare entailed—a small, traditional farm with animals experiencing a ‘natural’ outdoor environment. Within farmers’ narratives, it was evident that their pre-existing frames of animal welfare (e.g., ensuring resource needs are met) referenced what was important for welfare and thus formed the reference point from which positive welfare was judged. Consequently, positive welfare was assessed from the aspects of welfare farmers considered positive or good (e.g., maintaining health and well-being). Overall, farmers and citizens demonstrate two interpretation approaches whereby they looked to their linguistic and semantic associations with ‘positive’ and their pre-existing frames of animal welfare. This reflects the wider human decision-making literature, which notes that individuals often rely on what is most salient in the decision context (i.e., ‘positive’) and their previous experiences (i.e., existing animal welfare norms) to interpret ambiguous information (e.g., positive animal welfare) [50,51]. This has implications for the assimilation of positive welfare amongst individuals as such internal frames act as “mentally stored clusters of ideas that guide individuals’ processing of information” ([3], p.53). Consequently, as evidenced in this study’s findings, farmers and citizens did not approach positive welfare from a neutral state but assimilated it in a way which was consistent with their understanding of ‘positive’ and their own attitudes to farm animal welfare [52].

A particularly noteworthy consequence of the communication of ‘positive welfare’ is citizens’ association with ‘negative welfare’. This is all the more pertinent given the concerns many farmers raised about positive welfare producing such comparisons within society. As will be discussed in the following section, the knowledge of such internal frames reveals implications for the effective framing of positive welfare in communication.

## 5. Implications for the Framing of Positive Welfare in Communication

The internal frames individuals hold matter for how they interpret information and thus, how they act. As such, if animal welfare research is to be effectively transferred to key stakeholders, it is essential to consider what story or frame is engaged or reinforced by the message frames used to communicate it [53]. As suggested by this study’s findings, the term ‘positive animal welfare’ currently reinforces the comparison of negative and positive facets of animal welfare amongst citizens, and somewhat reinforces farmers’ perception that science and wider society is critical of their current welfare practices. As Entman (p.52, [3]) explains, “to frame is to select some aspects of a perceived reality and make them more salient in a communicating text, in such a way to promote a particular problem definition, causal interpretation, moral evaluation, and/or treatment recommendation for the item described”. Therefore, effectively framing positive welfare in communication requires a consideration of what aspects to make salient and how they will be received within the internal frames held by the target audience. The findings of this study highlight several implications in this regard.

First and foremost, effectively framing positive welfare in communication requires careful consideration of the salience of ‘positive’, and its indirect emphasis of ‘negative’, on individual perception. Citizens and farmers have limited opportunity or indeed inclination, to learn that positive animal welfare derives its name from human positive psychology. Consequently, its meaning is ambiguous, and direct reference to ‘positive’ engenders potentially detrimental framing effects; citizens make comparisons with negatives and farmers often defensively affirm their efficacy as competent animal caretakers. As it is not what is communicated but how something is communicated that matters, inclusion of the word ‘positive’ appears to disproportionately sway an individual’s thoughts to what ‘positive’ means to them in the context of animal welfare (e.g., extensive farming for citizens and good husbandry to farmers). Consequently, the essence of what positive welfare is trying to achieve (e.g., a balance of positive and negative experiences [20]) is lost in their cognition and assimilation of the term. As such, there is arguably a need to revolutionise the language used to communicate animal welfare. Rather than words that lead to comparative associations, effective message framing would make salient the overarching end goal of positive animal welfare (e.g., ‘animal well-being’ or ‘flourishing farm animals’) and as discussed below, consider the internal frames and motivations of the target audience.

For frames in communication to be effective, they should be “easily accessible and resonate with the existing beliefs of the audience” (p.3, [54]). However, this study found that ‘positive animal welfare’ is, unsurprisingly, unfamiliar to citizens and farmers, making it an ineffective communication frame. As such, it may be better to frame positive welfare in line with the target audience’s existing associations with farm animal welfare. For instance, citizens often associate good welfare with positive affective states [44], natural behaviours [55] and positive human–animal relationships [43]. Positive welfare supports and seeks to promote each of these factors [17]. Effective communication of positive welfare to citizens should thus make salient its association with these factors, so that these beliefs are communicated as applicable to their evaluation of positive welfare [9]. Framing positive welfare in line with farmers existing animal welfare beliefs may be even more critical, as farmers often resist external knowledge cultures [56,57] and make decisions in accordance with existing social and cultural capital (i.e., ‘the rules of the game’) [58]. As evident in both this study and previous research, farmers almost ubiquitously frame animal welfare in terms of health and productivity [45,46]. Thus, framing positive welfare in terms of its potential to enhance animal health and productivity [59] may prove effective, because it resonates with farmers’ existing beliefs. Additionally, as suggested by participating farmers, framing positive welfare’s potential to differentiate farmers on the market may be an effective communication strategy as it evokes farmers’ desire to convey their social and cultural capital as ‘good farmers’ to the external social world [58,60]. Using the existing belief frames of a target audience to frame a communicating message works, because it evokes the available information stored in the memory of the individual. This provides them with an appropriate point of reference from which to evaluate information, and thus supports more effective assimilation and understanding of the concept [9].

A further implication for effective communication framing is the underlying motivational orientation of the target audience. Research has found that an individual’s motivation to do something increases when that activity fits with their motivational orientation [61]. Central to this is the concept of regulatory focus, which posits that an individual can pursue the achievement of a goal with either a promotion-focused orientation or a prevention-focused orientation [62]. A promotion-focused individual “is concerned with advancement and accomplishment, with the presence and absence of positive outcomes” (p.171, [63]). Conversely, a prevention-focused individual is concerned with the “absence and presence of negative outcomes, with protection, safety and responsibilities” (p.171, [63]). For instance, farmers within the ‘proactive welfare improvement’ frame appear to hold a promotion-focus (e.g., want to advance and improve), while farmers in the ‘good husbandry’ frame appear to hold a prevention-focus (e.g., want to prevent negative outcomes or maintain current standards). Critically, leveraging the particular regulatory focus people are sensitive to can increase motivation [61]. Indeed, research finds that messages framed positively are more influential under a promotion-focus, while messages framed negatively are more influential under a prevention-focus [64]. This may account for why ‘good husbandry’ farmers did not frame positive welfare differently to pre-existing frames of welfare; its positive emphasis may not have matched with their prevention orientation. Conversely, ‘proactive welfare improvement’ farmers assimilated positive welfare as being something above and beyond existing welfare provisions, potentially as its positive emphasis matched their promotion orientation. This is supported by wider research findings that message effectiveness is enhanced when the message frame is compatible with the regulatory focus of the individual (e.g., prevention or promotion) [63,64,65]. Therefore, particular subsets and attributes of positive welfare (and indeed animal welfare more generally) may be best communicated in either prevention or promotion frames. For instance, health concerns may be best framed with a prevention focus, while enhancing positive affect in animals may be best framed with a promotion focus.

As Chong and Druckman (p.109, [9]) argue “frames in communication matter — that is, they affect the attitudes and behaviours of their audiences”. Thus, how positive welfare is communicated may determine how key animal welfare stakeholders understand and act in response. Yet, as noted in this study, ‘positive animal welfare’ represents an unknown and ambiguous concept to many. Consequently, the phrase may not effectively communicate what positive welfare is about or what it seeks to achieve. Moreover, the salience of the ‘positive’ appears to reinforce the undesired frame of negative welfare. To effectively frame positive welfare in communication, it is arguably better to reflect what positive welfare aims to achieve (e.g., a good quality of life, animal well-being, etc.) rather than where it originated (i.e., positive animal welfare reflecting the notion of positive psychology). Moreover, the effective communication and transfer of positive animal welfare from science to society may benefit from tailoring its message to the existing frames and motivations of the target audience. In this regard, a key question for welfare scientists is—what message should positive welfare convey and what behaviours should it motivate? This may require a deeper reflection on the vision for the future of positive welfare and the role of different societal actors within this.

## 6. Conclusions

Framing is an important theoretical framework for understanding why individuals interpret and perceive research evidence as they do and how best to communicate such research to different actors in the society. Positive animal welfare brings a change in frame to the traditional study of animal welfare, where greater emphasis is placed on the positive experiences in an animal’s life. However, little is known about how citizens and farmers may understand this concept or how such re-framing of welfare may impact their behavioural responses. This study was but an inaugural step into the exploration and examination of positive animal welfare in society and its effects on the perceptions, attitudes and behaviours of key animal welfare stakeholders. As no previous research, to the author’s knowledge, has asked or explored what ‘positive animal welfare’ means to citizens and farmers, it makes a fundamental contribution to the further development and expansion of positive animal welfare. Such explorations of the framing of positive welfare are timely, as positive welfare is a relatively recent concept that has not yet proliferated into wider society. This provides a great window of opportunity to ensure that positive welfare is developed in sympathy with the existing norms, values and meaning frames of key welfare stakeholders and is communicated in a way that resonates with and motivates, rather than alienates, such individuals. Understanding the internal frames individuals hold is essential for shaping the effective communication of positive welfare, and in turn, influencing the human behaviours which affect farm animal welfare.

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
