# Peer review of "Citizens’ and Farmers’ Framing of ‘Positive Animal Welfare’ and the Implications for Framing Positive Welfare in Communication"

_animals, 2019, doi:10.3390/ani9040147_

Round 1

Reviewer 1 Report

This paper studies the uptake of the idea of positive welfare among farmers and members of the general public. It builds on the assumption that the idea of positive welfare is clearly defined within animal welfare science: “… positive welfare research is no longer in its infancy and structures and frameworks for assessing positive welfare at the animal level are developing”. There is even according to the author a “positive animal welfare paradigm” with animal welfare science. The paper deals with “the transfer of positive welfare from science to society”.

However, this assumption is clearly false. Most references are to the work of David Mellor, but there are also references to a number of other authors discussing the idea of positive welfare. A careful reading of these references and others like them would reveal that there is a lot of confusion about what positive welfare is. 

Here are a few of the ideas found: a) Positive welfare is welfare taken to a higher level; b) positive welfare is about the occurrence of pleasure, play and other manifestations of positive mental states; c) positive welfare is about having a positive attitude, a good mood; d) positive welfare is about being occupied; e) positive welfare is about displaying species specific behaviour. And I absolutely sure that if the author did a study of the same kind here undertaken on a randomly selected group of animal welfare scientists results comparable to those here found will come out.

So the whole study is based on a false assumption regarding the existence of a positive animal welfare paradigm. For the study to be worth publishing it will be necessary at the outset to clarify some of the different views and meanings regarding positive animal welfare found among researchers working with this idea and then to discuss how the framings found within the two studied groups compare to these views and meanings. So in essence a new research question is required.

The study is based on qualitative interviews the results of which are interpreted in light of so-called on framing theory. This is OK, but particularly the demographic description of the citizen group is very thin. And one is left with the worry that not enough is done to secure diversity in terms of socio-demographic features. These limitations should be discussed. Also the author should abstain from making quantitative statements about (in %) about how many in the very small non-representative groups hold certain views. This is highly misleading.

Author Response

Dear Reviewer One,

I greatly appreciate your time in reviewing this paper and your support in its improvement.

Please find attached my point-by-point response to your recommendations.

WIth best regards,

Belinda Vigors

Reviewer 2 Report

Interesting and novel study aiming to uncover how farmers and citizens frame the concept of “positive animal welfare”. The study is well written, although I felt that several ideas were somehow repeated or too reiterative along the text, which makes the manuscript a bit long. This applies to most chapters, but particularly results and discussion. I suggest to shorten the text. Some examples are:

In the Introduction L111-112 is the same sentence as L 122-123.

In Methodology, the first 2 paragraphs are really part of an introduction. Moreover, here the author should concentrate on explaining how  the actual interview protocol was performed (L158 onwards) which is not clear enough. For instance, the methodology after having the recorded answers is not explained in detail (did the author have a protocol to extract the ideas on “positive” or “negative” animal welfare?); the number of interviews is rather small, particularly of citizens; if interviews were individual, how could the author have “diverse discussions between participants”??? (see L173).

In Results I suggest that figures 1 and 2 are not necessary, because they do not add anything new to data that are also expressed in the text.

In Discussion there are also several ideas that are repeated too much through the text, hence it could be shortened to be more concise. See for example L828 to 841.

Some minor precise comments follow:

L124….framers…should be farmers

L139…how individuals interpret and makes sense……make sense

L172….while the researcher remained a passive but active listener….(word missing after passive???)

L211….were detailed notes were taken……where

L504….husbandry frame were they…… where they…

L670….Consequently many farmers where perturbed…were perturbed

L729…..that a negative form does exists…….exists or does exist

Author Response

Dear Reviewer Two,

I greatly appreciate your time in reviewing this paper and your support in its improvement.

Please find attached my point-by-point response to your recommendations.

WIth best regards,

Belinda Vigors

Reviewer 3 Report

Please find my comments in the attached file

Author Response

Dear Reviewer Three,

I greatly appreciate your time in reviewing this paper and your support in its improvement.

Please find attached my point-by-point response to your recommendations.

WIth best regards,

Belinda Vigors

Reviewer 4 Report

This manuscript provides qualitative evidence of what relevant stakeholders mean when they are asked to describe 'positive animal welfare'. The author makes a good case in the intro for why this is important, and the study is well-justified. However, I have some concerns about the methods and results, as currently described, so I cannot recommend it for publication just yet.

Major points:

The method indicates that just one question was asked of the participants, but from this one question, a LOT of data appears to have been collected. Have I interpreted this correctly? Did the author really just ask one question, with no follow-ups at all, and then participants just rambled on and on? how long did each interview last? Please provide a range. Also, where did the interviews take place? By phone, over videoconference, in person? This information needs to be clarified because currently I have trouble understand how so much data were collected on the back of just one question.

The second major point is that the author states in several places (e.g. the abstract and the discussion) that stakeholders were 'inaccurate' in their descriptions of positive animal welfare, but she does not clearly define what is meant by positive animal welfare in this study, until the discussion at L885. That should be presented at the top of the intro, and, indeed, briefly in the abstract as well. But a greater concern about this part of the study is that the term 'positive animal welfare' uses a definition that was published about 8 years ago in an academic journal, and perhaps has not yet 'filtered' into the general public consciousness. Indeed, I am an animal welfare researcher and I myself would probably not have used this definition to explain what is meant by positive animal welfare. I don't think it's justified for the author to say that farmers and the general public describe the concept inaccurately if there's no widely accepted definition in the first place. I would update the discussion with a different interpretation of these results, which are interesting even without considering their 'accuracy'.

Finally, the results and discussion are both very long, and the discussion meanders a bit. I would recommend shortening the results considerably, perhaps combining some of the current themes where possible, and reducing the number of illustrative quotes. Same with the discussion; the author could cut probably a quarter of it without losing any impact. There is some repetition in both the results and discussion that are unnecessary and make it too lengthy.

Minor comments:

L20-21: provide a key example of two, rather than just saying the implications 'are further discussed'

L35: as in the above major comment - 'neither group wholly understand positive welfare' - by whose definition?

Keywords: just a tip - consider changing the first three. Words that are in the title will automatically be indexed, so the first three are redundant.

L124: I think the author means 'farmers'' instead of 'framers'

L170: 'facilitative' should be 'facilitate'

L172: 'passive but active listener' doesn't make sense to me. Please clarify

L225: in several places, there appears to be an unnecessary ', such as in this line when the authors writes 'farmers' and citizens' framed...'. This occurs throughout the text and is very minor, but a little distracting.

Figure 1 doesn't really add much; I would consider removing it.

L401: '...that probably does, it you know' is not clear. Either the punctuation is incorrect or there needs to be some clarification of the context.

Figure 2 is also probably unnecessary. There's no real need to add numbers to qual research like this. What about people who fall into more than one category? I don't think it's an instructive figure.

L783: 'extant' means 'surviving' or 'continuing to exist', not 'existing'. Consider replacing with 'existing'. Same for L904 in two places

L831 is a sentence fragment

Ref list is generally good, except for a few minor formatting discrepancies (e.g. a handful of journal names are spelled out, while the rest are abbreviated; some journal article titles only have the first word capitalised, while others capitalise each word).

Author Response

Dear Reviewer Four,

I greatly appreciate your time in reviewing this paper and your support in its improvement.

Please find attached my point-by-point response to your recommendations.

WIth best regards,

Belinda Vigors

Round 2

Reviewer 1 Report

I have now reviewed the revised manuscript; and I believe the

manuscript has been significantly improved and now warrants

publication in Animals.